# First Attempt at Synthetic Microbial Communities Design for Rearing Gnotobiotic Black Soldier Fly *Hermetia illucens* (Linnaeus) Larvae

**DOI:** 10.3390/insects16080851

**Published:** 2025-08-17

**Authors:** Laurence Auger, Marie-Hélène Deschamps, Grant Vandenberg, Nicolas Derome

**Affiliations:** 1Institut de Biologie Intégrative et des Systèmes, Université Laval, 1030 Avenue de la Médecine, Québec, QC G1V 0A6, Canada; nicolas.derome@bio.ulaval.ca; 2Département des Sciences Animales, Université Laval, 2425 Rue de l’Agriculture, Québec, QC G1V 0A6, Canada; marie-helene.deschamps@fsaa.ulaval.ca (M.-H.D.); grant.vandenberg@fsaa.ulaval.ca (G.V.)

**Keywords:** host-microbiota interactions, *Hermetia illucens*, edible insects, probiotics

## Abstract

Black soldier fly *Hermetia illucens* (Linnaeus, Diptera: Stratiomyidae) larvae are the most economically important edible insect worldwide, known for their efficient bioconversion of organic residues into organic fertiliser (frass) and larval biomass used in animal feed. The bioconversion process by the larvae is influenced not only by the insect’s natural capabilities but also by its microbiome, which is suggested to play a crucial role in larval growth and digestive functions. However, the effect of the microbiota has mainly been inferred from the analysis of microbiota available functions and not directly demonstrated. To explore the role of bacteria in larval growth, we designed and tested two synthetic bacterial communities (SynComs) and added these SynComs to either germ-free larvae (gnotobiotic) or conventionally reared larvae. These experiments were conducted on two distinct substrates, either vegetable-based or animal-based substrates. The SynComs increased gnotobiotic larval growth compared to untreated germ-free larvae reared in sterile conditions but not compared to conventional larvae reared in sterile substrate. In non-sterile conditions, SynComs improved growth on vegetable-based substrate but had no effect on animal-based substrate. These results highlight that the microbiota plays an important role in larval development and suggest that manipulating the microbiome could help to optimise insect farming.

## 1. Introduction

The edible insect sector has shown great potential in creating circular economies to reduce the environmental impacts of traditional farm agrifood systems, especially by rearing black soldier fly (BSF) *Hermetia illucens* on waste organic feedstocks meant for animal feed applications. BSF larvae are a particularly efficient up-cycler of organic residues, able to bioconvert a wide array of organic residues of animal and vegetal origin, even when reared on a diet substrate of poor nutritional value [1].

However, industrial rearing of BSF larvae must be profitable, putting pressure on the industry to optimise larval biomass production, occasionally to the detriment of the up-cycling advantages of the bioprocess. Organic residues that are locally available can vary significantly in composition and quality. The nutritional profile of BSF feed may be optimised by combining different organic waste streams or byproducts. However, availability may vary with regions and seasonal changes, and some of the most important sources of organic residues available may be of poor nutritional profile, which slows or reduces BSF larval growth (e.g., substrate with elevated lignocellulose content) [2].

Functional applications of microorganisms have shown promise to help resolve the challenge of poor nutritional rearing substrate [3]. The bacterial microbiota of BSF larvae is known to contribute significantly to host physiological functions, most notably in the digestive metabolic processes and growth. Owing to its essential contributions, the BSF microbiota has been the subject of numerous studies [4,5,6,7,8,9]. The dominant bacterial phyla in BSF are mostly consistent across studies. However, their relative abundance can vary greatly, as BSF shows a great capacity for metagenomic plasticity [10]. The BSF microbiota has been widely reported to be mainly influenced by the rearing substrate composition. In insects particularly, the microbiota functional repertoire specificity (i.e., the metagenome) enables host adaptation to ecological niches otherwise inaccessible, notably by compensating nutrient-poor diets [11]. Many insects with microbiota assembled from environmentally derived facultative symbionts have high intraspecific variability, which can be an important driver of adaptation [12].

Microorganisms are known for their ability to help their host to exploit its ecological niche by making resources in the environment available. Therefore, modulating the microbiota could enhance BSF growth performance. This is a promising strategy to tackle readily available substrates that are often nutritionally maladapted for BSF rearing.

Recently, probiotics (i.e., “live microorganisms that when administered in adequate amounts confer a health benefit to the host”) have been successfully used to this end [13]. For example, strains of *Bacillus subtilis* isolated from larvae were observed to increase prepupal mass [14]. Lignocellulolytic strains of *Bacillus* also accelerated biomass production and development time for larvae reared on dairy or chicken manure, potentially because the microbiota contributed to breaking down fibre present in a high ratio in these substrates [15].

Probiotics are also often promoted for their health benefits in farm animals, mostly to prevent diseases and infections. For example, *Pseudomonas fluorescens* and *Aeromonas sobria* have been used to prevent salmonid furunculosis in brook charr [16]. The BSF production sector has been able to boast about the lack of known disease affecting the larvae, but recently the Bacilli *Paenibacillus thiaminolyticus* has been identified as a larval pathogen inducing “soft rot” disease, which could cause massive losses in industrial rearing, where transmission between conspecific larvae is facilitated and difficult to prevent [17]. Until now, probiotics used in BSF research have mainly been selected based on probiotics with reported effects in other species models or selected by trial and error. Some studies have investigated the larval bacterial microbiota members that are culturable, providing a base for the selection of probiotic candidates [7,18,19]. However, the lack of knowledge of the microorganisms’ role in host functions is the main limitation to progress in this area of research. The roles ascribed to the BSF microbiota come from indirect studies or from functional inference [8,20].

Multi-strain microbial inocula used as potential probiotics have been investigated for their effectiveness in increasing the conversion rate, protein content, and survival rate of BSF larvae [21]. These studies have shown that microbiota can improve the performance of the larvae on its rearing substrate, highlighting the importance of the microbiota in BSF rearing systems. While it has been established that the microbiota are essential for larval development, it remains unclear which bacterial taxa play an essential role in the microbiota. To clarify the role of microbiota in BSF growth, we used a synthetic communities (SynComs) approach. Two SynComs were created from bacterial assemblies, chosen to resemble the previously characterised microbiota of BSF. The colonised BSF larvae were reared on two substrates, vegetable-based (VG) and animal-based (AN). The objective was to determine how differently assembled microbiota influence the growth performance of BSF, and if any microbiota member was essential for their growth. The SynComs were not optimised to be the most efficient for larval growth but are rather a first tentative approach to re-establish growth similar to conventionally reared larvae. We tested this ability to re-establish conventional growth by inoculating axenic larvae with the SynComs to create gnotobiotic organisms (i.e., an organism colonised with a specific community of known microorganisms). Monobiote (i.e., single organisms without any other microorganisms, referred to as “axenic” in this paper) larvae and conventionally reared larvae were used as controls. This study hypothesis is that BSF larvae need an assembly of bacterial taxa to reach the growth observed with naturally assembled microbiota. This study also proposes that single probiotics are insufficient to ensure the expected growth (i.e., mass and length) in the absence of environmental microbiota. Furthermore, the SynComs were projected to increase larval growth in conventional rearing conditions.

While other microorganisms than bacteria are abundant members of the microbiota, such as protozoa and fungi, they were not investigated in the scope of this paper.

## 2. Materials and Methods

**Black soldier fly.** Black soldier flies used in this experiment were taken from the rearing colony in *LAboratoire de Recherche en Sciences Environnementales et Médicales* (LARSEM) at Université Laval (Québec, QC, Canada). Re-established in 2020, this colony is continuously reared on Gainesville substrate and imagoes are inbred to reduce genetic variation [22]. The original colony as well as all experiments described in this study were performed under constant environmental conditions for the larvae of 30 °C ± 1 °C with 70% ± 5% relative humidity in a controlled environmental chamber (Conviron Ltd., Winnipeg, MB, Canada).

**Bacterial isolation.** BSF larvae were reared on two different diets: a vegetable-based diet (Gainesville) and an animal-based diet (poultry hatchery residues). Larvae were collected on days 4 and 6 post-hatching. Prior to extraction, larvae were surface-sterilised by washing with 70% EtOH, followed by rinsing with sterile water. The whole gut was dissected from day 6 larvae, as described in Auger et al., 2023 [23] For bacterial isolation, either dissected guts or whole larvae were pooled in groups of 3 and homogenised in 1 mL of tryptone soy broth (TSB, BBLTM Ref. 211768, BD, Mississauga, ON, Canada) using a sterile polypropylene pestle (DWK Life Sciences, Kimle, NJ, USA). The homogenate was then transferred into 9 mL of TSB and vortexed. A serial dilution approach was used to isolate bacteria: 100 μL of each dilution was added to cultures plates of MacConkey II Agar (BBLTM Ref. 212306, BD, Mississauga, ON, Canada), Starch M-Protein Agar (Himedia Ref. M801-500G, Kelton, PA, USA), Brain–Heart Infusion Agar (BBLTM Ref. 211065, BD, Mississauga, ON, Canada), TrypticaseTM Soy Broth with added 2% agar (i.e., TSA) (BBLTM Ref. 211768, BD, Mississauga, ON, Canada), and MRS (GranuCult^®^, Ref. 1.10660.0000, Frankfurter, Germany). Plates were incubated at 30 °C for 3 days (VWR^®^ B.O.D Refrigerated Peltier Incubator, VRI3P 89510-738, Avantor, Montréal, QC, Canada). Colonies with distinct morphology were picked with a sterile inoculation loop and re-streaked onto fresh culture plates repeatedly until cultures were pure (at least 3 times). Purity was verified by morphological observations under a phase contrast microscope. Pure cultures were preserved by inoculating 1 mL of TSA 15% *v*/*v* glycerol with a 24 h old CFU and stored at −80 °C. 

**Bacterial isolates identification.** DNA from each isolated bacteria was extracted following the method previously described [24]. The full-length 16S rRNA gene was then amplified with TaKaRa Taq DNA polymerase premix (Takara Bio USA Inc. R004A4, San Jose, CA, USA) following the manufacturer’s recommendations, using the primers F-tot and 1389-R. Amplicons were sequenced with the Sanger method at *La plateforme génomique of the Institut de Biologie Intégrative et des Systèmes* (IBIS) at Université Laval (Québec, QC, Canada). The quality score of the bases calls was verified on SnapGene, and poor quality (<40) ends were trimmed manually. A consensus sequence was generated using the BioEdit (version 7.7, https://bioedit.software.informer.com/) alignment function. Sequences with less than 100% identity were discarded. Consensus sequences were then classified with a BLAST (version 2.17.0) search in the NCBI nucleotide database.

**Selection of bacterial isolates for SynComs assembly.** Two SynComs were assembled, one using bacteria isolated from larvae reared on vegetable-based substrate and one from larvae reared on animal-based substrate. An arbitrary target of 8 bacteria was established for the assembly of each SynCom. Isolates were selected based on the following criteria: (1) the bacteria must be one of the four main phyla found in association with BSF; (2) the assembly must have the greatest taxonomic diversity possible; (3) known probiotics following previous criteria will be preferentially selected, (4) only bacterial isolates that formed distinct, countable colonies within 48 h of incubation were selected, to ensure both sufficient growth rate and colony clarity to establish a reliable growth curve. Standard growth curves were established for each candidate using optical density as described in [25].

**SynCom preparation.** The vegetable-based SynCom (SYVG) contained *Pediococcus pentosaceus*, *Weissella paramesenteroides*, *Enterococcus canintestini*, *Enterobacter hormaechei*, *Bacillus subtilis*, *Niallia circulans,* and *Acinetobacter radioresistens*. The animal-based SynCom (SYAN) included *Lysinibacillus macrolides*, *Paenibacillus xylenilyticus*, *Planococcaceae bacterium*, *Niallia circulans*, *Providencia rettgeri*, *Georgenia satyanarayanai*, and *Pseudomonas* sp. Between 48 and 24 h before the start of the experiment, one CFU from a fresh culture was used to inoculate 10 mL of TSB for each isolate. The concentration of the isolate culture was measured by the OD600 with the standard growth curve. SynComs were assembled directly into the experimental replicate container onto 400 g of diet, by adding each of the isolates at the desired concentration (e.g., for the treatment SYVG 5 × 10^7^, each isolate of the SYVG was added at a total of 5 × 10^7^ CFU on 400 g). The isolate *Pediococcus pentosaceus* was used as the individual probiotic treatment, a reported probiotic with effective antibacterial roles in microbial ecosystems, improving host homeostasis [26].

**Experimental treatments (list of conditions, rearing conditions, diet production).** For each experiment, treatments were performed in triplicate in individual transparent polypropylene Microbox containers of 5000 mL (SacO_2_, Deinze, Belgium) closed with a filter vented lid to allow air flow and prevent microbial contamination. Each container was sterilised separately by autoclave before use. Two rearing substrates were used in this experiment: vegetable-based substrate (VG) and animal-based substrate (AN). The VG substrate was designed to represent food-waste from local grocery stores; and it was composed of 39% of fruits (1% lemon, 2% strawberry, 2% grape, 2% pear, 3% pineapple, 4% apple, 4% banana, 4% cantaloupe, 5% tomato, 5% orange, and 7% bell pepper), 36% of vegetables (2% onion, 2% cauliflower, 2% sweetcorn, 3% green cabbage, 3% carrot, 3% leak, 3% celery, 3% broccoli, 5% potato, and 10% romaine lettuce), 10% spent distiller’s grains, and 15% bread [27]. Ingredients were purchased in July 2022 from a local supplier (Tout Prêt Inc., Québec city, QC, Canada) and shredded in a homogenous substrate with an industrial food processor (Rietz disintegrator, model: RA2-8-K322; Bepex International LLC, Minneapolis, MN, USA). The AN was composed of hatchery residues provided by a local producer (Couvoir Scott, Saint-Apollinaire, QC, Canada) [28]. This substrate was a homogenous mix of eggs, chicks, and other production residues ground into a fine paste, with 30% chicken manure added as a structuring agent. Both substrates were adjusted to 70% humidity on a dried basis with distilled water. Substrates were stored at −20 °C in batch of 1 kg in vacuumed sealed bags until used. The VG substrate was either sterilised or not (conventional). Larvae added to the VG substrates were either axenic or not (conventional). These combinations gave 4 initial conditions: (1) sterile VG substrate with axenic larvae (sterile diet, sterile larvae; abbreviated to VG SF SL), (2) non-sterile VG substrate with conventional larvae (non-sterile diet, non-axenic larvae abbreviated to VG NF NL), (3) sterile VG substrate with non-axenic larvae (sterile diet, non-axenic larvae abbreviated to VG SF NL), (4) non-sterile AN substrate with non-axenic larvae (non-sterile diet, non-axenic larvae, abbreviated to AN). A total of 6 treatments were tested: (1) bacteria isolated from the larvae and potential probiotic *Pediococus pentosaceus* at 5 × 10^7^ CFU (G01) [26]{Citation}, (2) VG SynCom at 5 × 10^7^ CFU/isolate (SYVG 5 × 10^7^), (3) AN SynCom at 5 × 10^7^ CFU/isolate (SYAN 5 × 10^7^), (4) VG SynCom at 10^8^ CFU/isolate (SYVG 10^8^), (5) AN SynCom at 10^8^ CFU/isolate (SYAN 10^8^), (6) both SynComs combined at 10^8^ CFU/isolate (SYVGAN 10^8^). Larvae that were axenic at the start of the experiment and were then treated by the addition of one or multiple microorganisms are thereafter referred to as gnotobiotic larvae.

For each treatment, 400 g of either the VG or AN rearing substrate was added (at 70% RH) in the rearing container. Then each of the isolates composing the SynCom were added at either (1) 5 × 10^7^ CFU (final concentration of 1.25 × 10^5^ CFU/g for each isolate) or (2) 10^8^ CFU (final concentration of 2.5 × 10^5^ CFU/g for each isolate) and thoroughly mixed with a flame-sterilised spoon, before adding 400 neonates (see SynComs preparation and sterile condition).

Due to limitations in time and resources, it was not possible to conduct all experimental conditions in a single experiment. Therefore, the study was divided into two separate experiments. In each experiment, control conditions and some experimental conditions were repeated to ensure reliability. Additionally, each condition within each experiment was performed in triplicate to maintain consistency and accuracy.

**Sterile condition and axenic larvae.** The VG substrate was sterilised by autoclaving 3 times for 30 min at 24 h intervals in batches of 100 g placed in 500 mL glass jars covered with sterilisation wraps taped in place (Propper Steri-Wrap™, Thermo Fisher Scientific 11-890-8A, West Etobicoke, ON, Canada). Egg clutches were collected from cardboard strips, and their surface was sterilised as described in Auger et al., 2023 [20]. Sterilised eggs in clutches of 4 to 5 were left to hatch in a sterile ventilated flask with 200 mL of TSB. After hatching, only flasks without visible microorganism growth were used as axenic larvae. Around 400 neonates were transferred to each experimental replicate. All the manipulations for sterile conditions and sterile larvae were performed under laminar air flow, with flame-sterilised instruments between each manipulation. Non-axenic (conventional) neonates were obtained using the same approach as for the sterile condition but washes and TSB were replaced by PBS 1X physiological solution (137 mM NaCl, 2.7 mM KCl, 10 mM Na_2_HPO_4_, 1.8 mM KH_2_PO_4_, pH 7.0).

**Sterile conditions control.** To assure the sterilised substrate was truly sterile at the start of the experiment, each batch of sterilised substrates was sampled for 1 g after thoroughly mixing with a flame-sterilised spoon under laminar air flow, and the sample was suspended in 10 mL of TSB in a 15 mL culture tube. The culture was kept for 7 days at 30 °C in aerobic conditions. If no growth was observed after one week, sterility was confirmed, otherwise the replicate was removed from the experiment. All the manipulations with sterile diet were performed under laminar air flow, with flame-sterilised instruments between each manipulation.

**Sampling process and phenotypic measurements.** Samples for bacterial taxonomic analysis were taken at days 4, 8, and 12 post-hatching (hatching = day 1). A sample of 3 pooled larvae was taken randomly from each replicate, and flash-frozen in liquid nitrogen before storing at −80 °C until RNA extraction. Simultaneously, 1 mL of substrate was sampled after thorough mixing and stored at −80 °C until RNA extraction. All sampling instruments were sterilised with 70% ethanol and a flame between each replicate. Phenotypic measurements were taken on 10 larvae sampled at random from each treatment replicate on days 4, 6, 8, 10, and 12 post-hatching. Larvae that could not be immediately measured were kept at −20 °C until analysis. Individual length was measured using a digital calliper (live larvae were put on ice in a Petri dish), and mass was measured with a precision balance (±0.0001 g, Mettler ToledoTM Standard ME Analytical Lab Balance, Mississauga, ON, Canada). Larval mass and length was compared between treatments by ANOVA with larval age as a within-subject factor and the treatment as a fixed factor. Multiple comparisons were conducted using the Tukey HSD method, using a significance threshold of *p* < 0.05.

**RNA extraction, library preparation, and sequencing.** RNA extraction from frozen larvae was performed by adding 1 mL of TRIZol reagent (Invitrogen™ 15596018, Life Technologies, Burlington, ON, Canada) heated for 5 min at 50 °C, before using on the frozen larval samples. The larvae were then homogenised using a sterile RNase-Free polypropylene pestle (DWK Life Sciences, Kimle^®^, Wertheim, Germany). The following RNA extraction steps proceeded according to the manufacturer’s instructions. DNA synthesis was conducted using qScript^®^ cDNA SuperMix (QIAGEN, Toronto, ON, Canada) according to the manufacturer’s instructions. Targeted 16S rRNA gene V3-V4 regions (500 pb) were amplified by PCR using Q5 high-fidelity DNA polymerase (New England BioLabs, Inc., Ipswich, MA, USA) with the forward primer 341F (5′–ACACTCTTTCCCTACACGACGCTCTTCCGATCTCCTACGGGRSGCAGCAG–3′) and the reverse primer 785R (5′–ACACTCTTTCCCTACACGACGCTCTTCCGATCTGACTACHVGGGTATCTAATCC–3′). The PCR amplification program was (1) 30 s DNA denaturation at 98 °C; (2) 35 cycles amplification steps of 10 s (denaturation) at 98 °C, 30 s (annealing) at 64 °C and 20 s (elongation) at 72 °C, and a final elongation step; (3) 2 min at 72 °C. The post-PCR DNA concentration and quality were assessed by Nanodrop and by electrophoresis on 2% agarose gels, and poor-quality samples were discarded. Amplified DNA was purified with magnetic beads (AMPure beads, Beckman Coulter Genomics^®^, Mississauga, Canada) and sequenced on a MiSeq platform from Illumina by *La plateforme génomique at Institut de Biologie Intégrative et des Systèmes* at Université Laval (Québec, QC, Canada). All of the sample sequencing data is available on NCBI Sequence Read Archive (SRA) in the project PRJNA1282731.

**Processing of data from 16S rRNA gene sequencing and statistical analysis.** Sequence analysis was performed using the R (version 4.3.3) and Rstudio programs (version 4.2.1) [29,30]. Reads were timed to remove poor-quality reads (phred < 20) and those truncated at the extremities (280 bp forward and 240 bp reverse cut-off). Short reads (under 420 bp) were removed. Taxonomic assignment was performed using SILVA 138 SSU database. PCR negative controls were used to remove potential contamination in the dataset with the prevalence method from the decontam package (version 3.6) using default values [31]. Reads matching chloroplast and mitochondrial sequences were filtered out. Samples with fewer than 500 ASVs were discarded as quality control since they had very low abundance (<0.016% compared to the sample mean), except for samples from the sterile diet conditions. The Phyloseq package (version 3.16) and ggplot2 packages (version 3.5.0) were used to produce relative abundance graphics, and rare ASVs were removed to denoise the graphics (<1 in at least 10% of samples) [32,33]. Alpha diversity (bacterial microbiota richness and evenness) was measured with the alpha-diversity Pielou and Shannon entropy indexes. Statistical tests were performed with multivariate ANOVAs followed by Tukey’s honestly significant difference tests (Tukey’s HSD) when normality was confirmed by the Shapiro test. If not, Kruskal–Wallis test and pairwise Wilcoxon test were employed. Beta diversity (bacterial microbiota compositional diversity) was measured using the vegdist function of the VEGAN package (version 2.6-8) [34]. The effect of the treatment on the microbiota profile was measured using the adonis function by permutational multivariate analysis of variance (PERMANOVA) based on the massed UniFrac distances with 1000 permutations. All statistical analysis were conducted using a significance threshold of *p* < 0.05.

## 3. Results

### 3.1. Bacterial Isolates

In the 50 isolates from VG-reared larvae that were sequenced, 13 unique bacterial species were identified, while 12 unique bacterial species were found in the 46 isolates from AN-reared larvae (Table 1). The only isolate identified in common for both substrates was *Niallia circulans.*

### 3.2. Individual Larval Mass and Length

The measurement of larval mass and length are presented for each experimental treatment in Figure 1. The effect of treatments on larval growth was time-dependent. For sterilised larvae that were reared on non-sterilised VG substrate (VG-NF-SL-CTRL day 12 mean mass = 0.21 ± 0.01 g), their final mass and length was similar to those of conventional larvae reared on the non-sterilised VG substrate (VG-NF-NL-CTRL, day 12 mean mass = 0.22 ± 0.01 g) (Figure 1a EXP2), and both had a significantly greater final mass and length than axenic larvae reared on sterilised VG substrate (SF-SL-CTRL, day 12 mean mass = 0.005 ± 0.001 g). Axenic larvae reared on sterilised VG and treated with the VG SynCom at 5 × 10^7^ CFU (VG-SF-SL-SYVG 5 × 10^7^) had the greatest mass and length gain of all gnotobiotic treatments in the first experiment (Figure 1a EXP1). Larvae treated with the single probiotic candidate (VG-SF-SL-G01) had greater growth than axenic larvae.

In the second experiment (Figure 1a EXP2), the axenic larvae treated with the three VG SynComs (VG-SF-SL-SYVG 5 × 10^7^, VG-SF-SL-SYVG 10^8^ CFU, and VG-SF-SL-SYVGAN 10^8^ CFU) all had a similar final mass, which was significantly greater than that of the untreated sterilised larvae (VG-SF-SL-CTRL), but lower than the larvae reared on non-sterilised VG substrate (VG-NF-NL-CTRL and VG-NF-SL-CTRL). The AN SynCom did not significantly change the final mass of the larvae (VG-SF-SL-SYAN 10^8^ CFU, day 12 mean mass = 0.017 ± 0.011 g) compared to the sterile control (VG-SF-SL-CTRL).

Larvae reared on non-sterilised VG substrate (Figure 1b EXP2) treated with the combined VG and AN SynComs at 10^8^ CFU (VG-NF-NL-SYVGAN 10^8^ CFU) had a greater final larval mass and length compared to all the other treatments and control. No other treatment differed from the control. No significant difference was observed for the final mass and length between experiments for the control larvae reared in the NF NL VG condition (Figure 1b).

In the AN condition, SYAN was the best-performing treatment (mean mass = 0.14 ± 0.05 g, *p* < 0.05), while SYVG treatment had the lowest final mean mass (0.16 ± 0.03 g, *p* > 0.05) but did not differ from the control.

### 3.3. Active Microbiota Composition

The final samples used for microbiological analysis after quality filtering for each condition are presented in appendix Table A1. There were a total of 237,757 ASVs, a mean of 31,608 ASVs by sample (median of 30,170 ASVs), and a minimum of 612 ASVs and a maximum of 110,393 ASVs, excluding samples from the axenic group. When including samples from the axenic group, there was a minimum of 1 ASV per sample, a mean of 28,830 ASVs, and a median of 27,930 ASVs. Most of the dataset was composed of rare taxa, with 75,375 ASVs undetected in at least 10% of all samples. Several samples were discarded, due to poor cDNA and/or read quality, including rearing substrate and larval samples taken from the VG-SF-SL-CTRL condition (Table A1). Only samples from one specific replicate of the VG-SF-SL-CTRL condition from the first experiment passed the filtering process and revealed a high level of contamination. Therefore, these samples were discarded from the analysis, and the sample from this replicate was removed from the phenotype analysis. The relative activity of the bacterial microbiota taxa in the BSF larvae reared on VG substrate is presented for each treatment in Figure 2. BSF microbiota activity in VG SF condition was almost exclusively carried out by Bacillota and Pseudomonadota. The larvae reared in the AN condition (Figure 3) had a distinct bacterial activity profile, with Pseudomonadota and Bacteroidota as the dominant phyla, and to a lesser degree Bacillota. The bacterial taxa in the rearing substrates are presented by their relative activity in Figure 3. The most active phyla and class taxa found in the rearing substrate were similar to those of the larvae (Figure A1, Figure A2 and Figure A3).

### 3.4. Rearing Substrate Effect

The larvae and the substrate microbiota richness and evenness were significantly (*p* < 0.05) modulated by the rearing substrate (Figure 4 and Figure A1). The BSF larvae reared in the AN condition had the highest microbiota richness and evenness for all time points, followed by the VG NF substrate. As expected, the VG SF substrate had the lowest microbiota alpha diversity. The substrate was also the main factor modulating the microbiota composition (beta diversity) as presented in Figure 5. The sterilisation process of the eggs leading to the removal of the surface microbiota was not found to affect the bacterial structure of the larvae’s microbiota reared in the NF VG condition, except for the last sampling time (Figure 6). 

## 4. Discussion

This study developed two SynComs from bacteria isolated from BSF larvae reared on two substrates and proceeded to investigate the effect of these SynComs on larval growth and the taxonomic distribution of their active microbiota. The treatment with the greatest effect on larval final mass for all conditions was the combination of the two SynComs (SYVGAN 10^8^ CFU) administered to larvae reared on non-sterilised VG (Figure 1b). The larval final mass and length differed for the same treatment between experiments 1 and 2 (Figure 1a). This variation may have been caused by the lower relative humidity observed in the second experiment. The second experiment was performed in the same environmental unit (Conviron) as the previous experiments, using the same settings (70% RH, 12:12 L:D, 30 °C); however, independent instruments (hydrometer Honeywell H10C) used for the measure of RH showed the humidity to be at 50 ± 5% RH in the second experiment. A lower relative humidity is associated with a slower developmental rate in BSF [35].

**Larval growth is dependent on active microbiota composition.** Sterilisation of the VG substrate was associated with a lower growth rate in all conditions. The mass gain efficiency of the BSF larvae appears to primarily result from the microbiota assembled from environmental bacterial strains. This plastic microbiota likely offers metagenome functions that are able to efficiently transform available nutrients into biomass. It could be hypothesised that the added bacteria biomass from the treatments provides an additional food source for the larvae, enhancing larval growth. However, the initial dosage of the individual SynComs did not impact larval growth performance, even under sterile conditions. Further testing of higher dosage of administered treatments may show an effect on growth performance, and direct observation of bacterial digestion by the larvae would be needed to better investigate this hypothesis. The best-performing treatment was the one with the highest bacterial diversity (SYVGAN). This could indicate that the complexity of the microbiota used as treatment may be the key to enhancing the BSF larvae rearing performance. SYVGAN was the only treatment to increase larval mass in the NS VG condition, likely because it offered the greatest diversity of bacterial taxa, potentially translating into the greatest functional diversity. In humans, reduced microbiota diversity was associated with reduced growth [36]. While this study focuses on the growth of the larvae, a diverse microbiota also contributes to many other functions of the host, such as hormones regulations, host physiology, immune defence, and protection against pathogens. Therefore, a more diverse assembly may offer a wider range of functions or synergetic processes to break down complex carbohydrates and proteins, making more resources available to the host, much like generalist feeders are associated with higher microbiota diversity than specialists [37]. BSF larvae’s bioconversion efficiency may depend on the presence of a group of specific functions in the metagenome, and the need for the specific functions is probably dictated by the rearing substrate. This would explain why the microbiota composition of BSF is dependent on the rearing substrate [10]. This could indicate that the “core microbiota” needed by BSF is not the result of close association with bacterial symbionts, but a highly dynamic assembly reflecting the needs of BSF to exploit a specific rearing environment [37].

**Functions of the SynCom isolates.** Administered treatment and dosage altered the taxonomic distribution of active taxa in larval microbiota and the environmental microbiota (Figure 5). The main roles of the bacterial gut microbiota in insects are amino acid biosynthesis, protein, lipid, and carbohydrate digestion, energy metabolism, and biosynthesis of vitamins [38]. The lower relative abundance of active Bacillota in AN-reared larvae was surprising, as the functional role of this phylum in BSF larvae has previously been linked to the digestion of animal products [39]. Many species of Bacteroidota are degraders of high-molecular-mass organic matter, with many species identified as cellulolytic, like the *Bacterioides cellulosilyticus*. Bacteroidota are therefore suited to contribute to the digestion of BSF larvae reared on vegetable-based substrates, explaining their high abundance in the larvae from the VG condition. A key difference observed in the bacterial microbiota was the high activity of the Actinomycetota phylum in larvae reared on non-sterilised substrate compared to those on the sterilised substrate (Figure 2). Actinomycetota are commonly associated with multiple protective symbiotic interactions with insect host species [40]. Members of this phylum are suspected to have little involvement in host digestive functions despite their ability to exploit carbon from a wide variety of sources [41]. Despite their expected importance in insects, they were also almost completely absent in the AN substrate and associated larvae, as well as in the larvae reared on SF VG, although they were one of the isolates included in SYAN (Figure 2 and Figure 3). In the AN substrate, we identified the activity of the potential pathogens *Pseudomonas* sp., *Clostridium botulinum*, *Streptococcus pseudopneumoniae,* and *Klebsiella pneumoniae*. Animal residues like the AN substrate harbour high bacterial density, with little niche left to colonise by exogenous bacteria. The high diversity and density of this substrate may provide a resistance to the colonisation of exogenous bacteria genera, which could be a limitation to the use of probiotics and SynComs. A previous study has suggested that the high relative abundance of Pseudomonadota may be a potential microbial signature of disease in BSF as has been suggested in humans [42]. Pseudomonadota are the most phenotypically diverse phylum in the Bacteria domain. The functional diversity of the phylum and the results of the present study rather indicate that the activity of Pseudomonadota in the larval microbiota cannot be used as an indicator of disease in BSF. Furthermore, the prevalence of disease in BSF has not been investigated in relation to microbiota composition.

The rearing substrate was the main factor responsible for the variance in bacterial composition and alpha diversity (Figure 4 and Figure 5). SynComs appeared to perform better when they were used on the same substrate from which they were originally isolated. The SYVGAN treatment had the most positive effect in larvae reared in VG substrates, while SYAN was the worst-performing treatment. This may be because the SYAN assembly lacks bacterial function in its metagenome that contributes to host digestion. This assembly could also provoke a rise in competitive interactions by introducing exogenous and potentially opportunistic taxa in the condition. The resulting gut dysbiosis may temporarily slow the larval growth. In the AN condition, SYAN was the best-performing treatment, while SYVG was the worst-performing treatment. This may be indicative that SYAN was also better suited to its substrate of origin. These results point toward a substrate-dependent effect of the SynCom used, since the bacterial taxa isolated from an environment and used for SynCom assembly are expected to be adapted to exploitation of the environment they came from.

The surface sterilisation of the eggs to produce axenic larvae did not influence the final larval mass and length compared to untreated larvae, nor did it influence the bacterial community taxonomic composition. These results support previous reports that BSF external mechanisms of microbiota vertical transfers have a limited contribution to BSF larval bacterial microbiota composition [23].

There is currently a gap in the knowledge regarding what makes a bacterial inoculant beneficial to the desired host phenotype. Most studies use a guesswork approach based on the efficiency of the inoculant in other host systems. To further advance the study of BSF microbiota engineering, it is essential to determine whether the metagenomic plasticity is due to recruitment of environmental taxa as a key element for substrate adaptation. This highlights the necessity to functionally characterise BSF bacterial isolates to enhance SynCom efficiency. With a functional profile of the microbiota, SynComs could be assembled according to the contribution of the metagenome to the host metabolic functions, instead of being simply based on bacterial taxa occurrence.

This study had no way to track the establishment of the administered strains in the larval microbiota, which is a factor needed to characterise probiotics. This is because a relatively long-term interaction between the probiotics and the host is expected to be necessary for the host to receive a beneficial effect from the microorganisms. However, because insects like larvae live in very close proximity with a specific ecological niche, the colonisation of the rearing substrate may also provide close interaction between the administered microorganisms and the host, leading to beneficial outcomes. We would suggest for future studies to investigate the persistence of candidate probiotics in the larval gut and the rearing substrate using approaches like fluorescent in situ hybridisation [43]. A limitation of this study was the limited number of isolates in the candidate pool from which the SynComs were created, as well as the selection of fast-growing isolates to prepare the assembly, meaning some interesting candidates like *Levilactobacillus brevis* were not considered. Investigating the role of these microorganisms in BSF larval growth and development could clarify whether they play an essential role that would benefit a new SynCom design. Furthermore, no anaerobic bacteria were isolated and included in the assembly because the substrate environment is known to harbour many pathogens, including *Clostridia botulinum*. The culture of both opportunistic and anaerobic bacteria would have necessitated a level 2 laboratory and complicated the inoculation process.

## 5. Conclusions

The results presented in this study suggest that the main factor contributing to the BSF larval growth performance (length and mass gain) may be the diversity of the bacterial communities assembled in its microbiota and the adaptation of this assembly to the rearing substrate. The SynComs developed in this study were arbitrary microbial assemblies that would benefit from further refinement. To optimise SynCom design, finer tuning of isolate selection is crucial, focusing not only on individual probiotic effect but also on intra-microbial interactions and the adaptation of the functional repertoire to the rearing substrate. This would enhance synergy and stability within the community, promoting more beneficial outcomes. The positive impact of SYVGAN on larval growth showed the potential SynComs could offer for controlling the bias of microbiota variability across studies by developing gnotobiotic larvae while still achieving similar results to conventional rearing systems. Harnessing the functional diversity of the microbiota may be the key to BSF growth performance. The great metagenome plasticity of the bacterial microbiota in BSF may be hiding a functional redundancy of the healthy microbiome for a given substrate, similar to what has been observed in humans [44].

## Figures and Tables

**Figure 1 insects-16-00851-f001:**
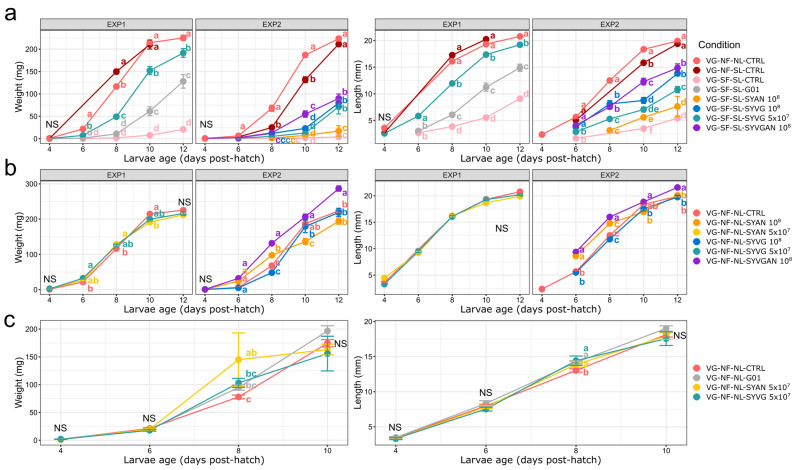
Treatment effect on larval mean mass and mean length. Mean mass and length of larvae from each treatment (triplicates, n = 30): control (CTRL), single probiotic candidate (G01), VG SynCom at 5 × 10^7^ CFU/isolate (SYVG 5 × 10^7^), AN SynCom at 5 × 10^7^ CFU/isolate (SYAN 5 × 10^7^), VG SynCom at 10^8^ CFU/isolate (SYVG 10^8^), AN SynCom at 10^8^ CFU/isolate (SYAN 10^8^), and both VG and AN SynComs combined at 10^8^ CFU/isolate (SYVGAN 10^8^). Conditions are presented: (**a**) axenic larvae (prior to treatment, SL) reared on sterilised vegetable-based substrate (SF VG), (**b**) larvae (NL) reared on non-sterilised vegetable-based substrate (NS VG), and (**c**) larvae (NL) reared on animal-based (AN) substrate. Statistical differences in mean mass and mean length between treatments were computed by ANOVA between treatments by day (*p* < 0.05). A different letter indicates statistical significance.

**Figure 2 insects-16-00851-f002:**
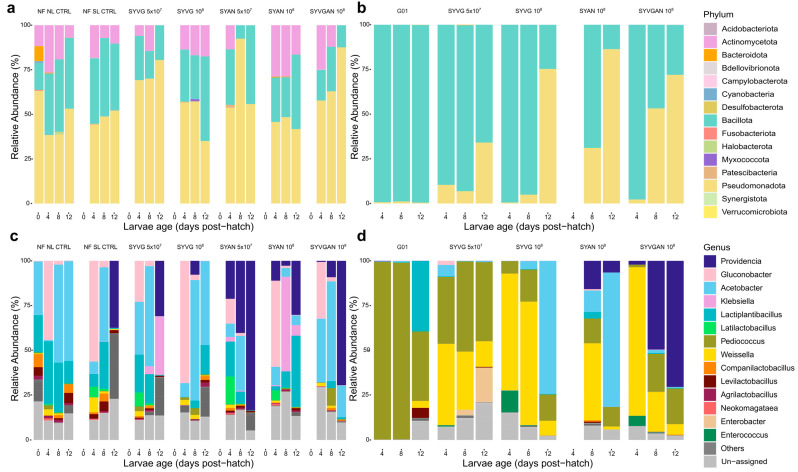
Relative abundance of active bacterial microbiota in BSF larvae reared on the VG substrate. Most abundant active phylum bacterial taxa are shown for (**a**) the larvae (NL) reared on the non-sterilised substrate (NF) and (**b**) the gnotobiotic larvae (SL) reared on the sterilised substrate (SF). The bacteria genera are presented for (**c**) the non-sterilised substrate (NF) and (**d**) the sterilised substrate (SF). The substrate bacterial activity in the VG substrate before the treatments and the introduction of BSL is presented on day 0 for NF NL CTRL. Taxa with a relative abundance lower than 1% in all samples are indicated as “others”.

**Figure 3 insects-16-00851-f003:**
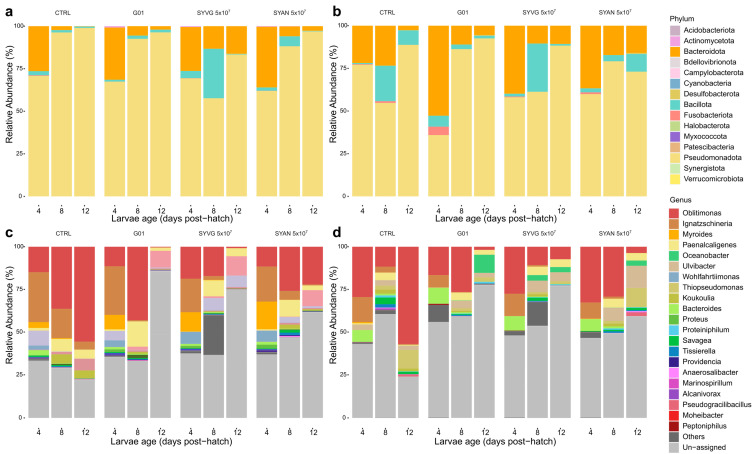
Relative abundance of active bacterial microbiota in the larvae and the AN rearing substrate. The most abundant active bacterial Phylum taxa are presented for (**a**) the animal-based substrate (AN) larval samples and (**b**) the substrate samples. The genera are shown for (**c**) the animal-based substrate (AN) larval samples and (**d**) the substrate samples. Taxa with a relative abundance lower than 1% in all samples are indicated as “others”.

**Figure 4 insects-16-00851-f004:**
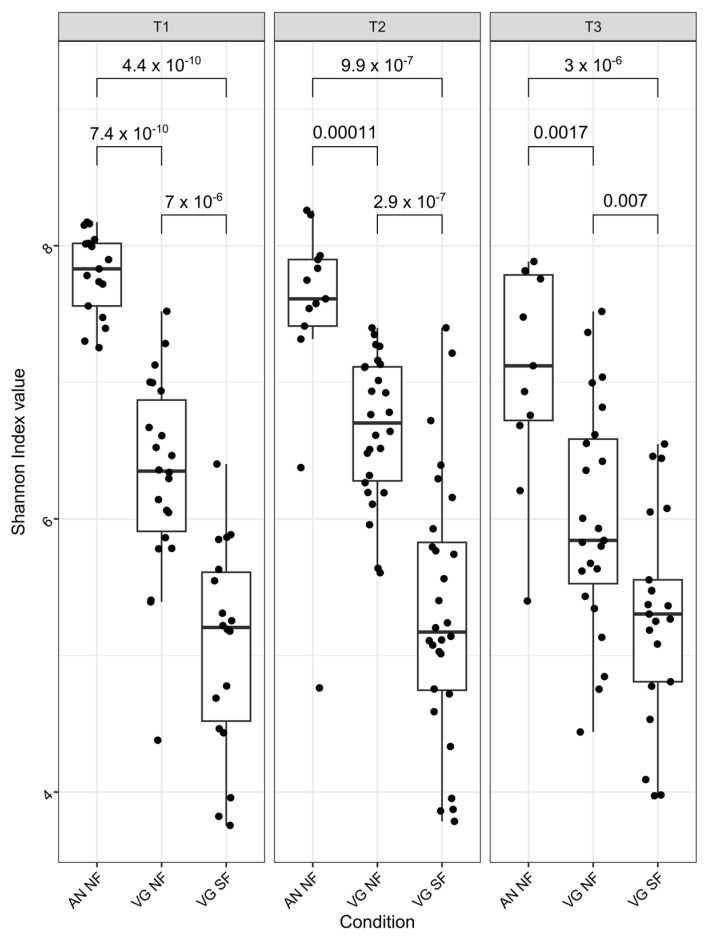
Effect of the rearing condition on the larval active bacterial microbiota diversity. Shannon–Wiener index values are presented for the bacterial microbiota of all the larvae reared in the animal-based substrate (AN FN), the non-sterilised vegetable-based substrate (VG NF), and the sterilised vegetable-based substrate, at each of the three sampling times (day 4, 8 and 12).

**Figure 5 insects-16-00851-f005:**
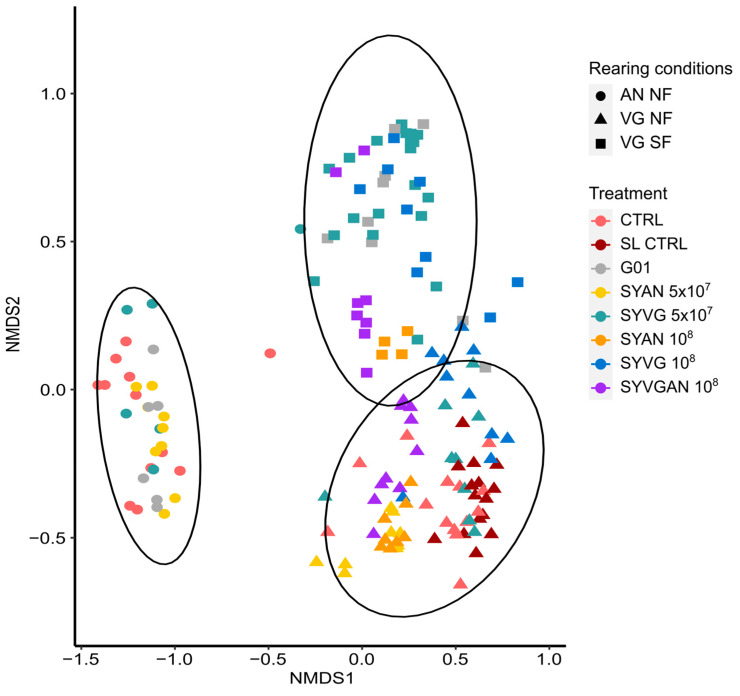
Rearing condition effect on the taxonomic distribution of active bacterial microbiota. Non-metric multidimensional scaling (NMDS) with group distance comparison using PERMANOVA showed significant differences between the active bacterial microbiota of BSF larvae based on rearing conditions: animal-based substrate (AN NF), vegetable-based non-sterilised substrate (VG NF), and vegetable-based sterilised substrate (VG SF). Treatments are control (CTRL), single probiotic candidate (G01), VG SynCom at 5 × 10^7^ CFU/isolate (SYVG 5 × 10^7^), AN SynCom at 5 × 10^7^ CFU/isolate (SYAN 5 × 10^7^), VG SynCom at 10^8^ CFU/isolate (SYVG 10^8^), AN SynCom at 10^8^ CFU/isolate (SYAN 10^8^), and both VG and AN SynComs combined at 10^8^ CFU/isolate (SYVGAN 10^8^).

**Figure 6 insects-16-00851-f006:**
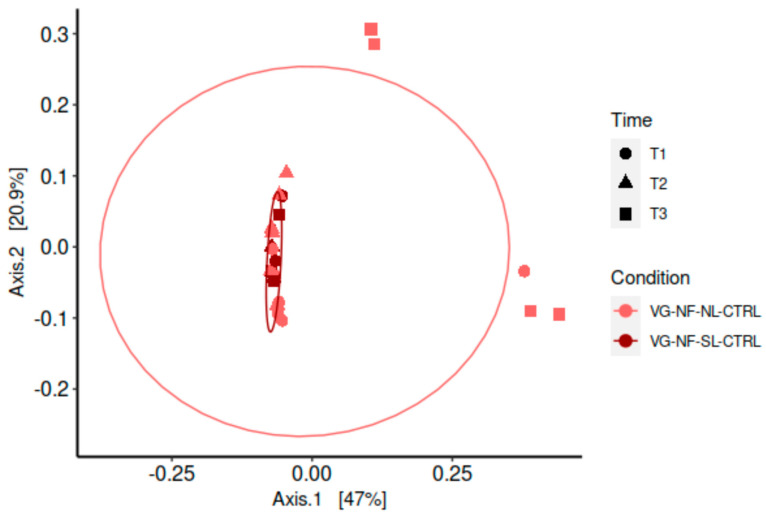
Effect of egg surface sterilisation on the taxonomic distribution of active bacterial microbiota. PCoA with group distance comparison using PERMANOVA showed no significant differences between larval microbiota composition between larvae from surface sterilized eggs or untreated eggs reared in the NF VG condition, except at the latest sampling time (day 12 post-hatch). Samples (n=6/condition) were taken at three times (day 4, 8 and 12 post-hatch).

**Table 1 insects-16-00851-t001:** Classification of bacteria isolated from black soldier fly larvae.

Isolate Origin	Phylum	Class	Genus	Species	AC
Larvae reared on vegetable-based diet	Pseudomonadota	Gammaproteobacteria	*Proteus*	*mirabilis*	CP053718.1
	Pseudomonadota	Gammaproteobacteria	*Enterobacter*	*hormaechei*	CP045611.1
	Pseudomonadota	Gammaproteobacteria	*Klebsiella*	*pneumoniae*	OW995946.1
	Bacillota	Bacilli	*Pediococcus*	*pentosaceus*	MT604839.1
	Bacillota	Bacilli	*Levilactobacillus*	*brevis*	ON724232.1
	Bacillota	Bacilli	*Weissella*	*paramesenteroides*	MT613524.1
	Bacillota	Bacilli	*Enterococcus*	*canintestini*	MF092811.1
	Pseudomonadota	Gammaproteobacteria	*Enterobacter*	*cloacae complex (hormaechei)*	CP045611.1
	Bacillota	Bacilli	*Enterococcus*	*gallinarum*	MT597704.1
	Pseudomonadota	Gammaproteobacteria	*Klebsiella*	sp.	KY689936.1
	Bacillotas	Bacilli	*Bacillus*	*subtilis*	ON795918.1
	Bacillota	Bacilli	*Niallia*	*circulans*	CP053989.1
	Pseudomonadota	Gammaproteobacteria	*Acinetobacter*	*radioresistens*	MT367790.1
Larvae reared on animal-based diet	Bacillota	Bacilli	*Paenibacillus*	*xylanilyticus*	CP044310.1
	Pseudomonadota	Gammaproteobacteria	*Providencia*	*rettgeri*	LT899977.1
	Bacillota	Bacilli	*Planococcaceae*	*bacterium Storch*	KX957962.1
	Bacillota	Bacilli	*Lysinibacillus*	sp. *(louembei)*	ON520837.1
	Bacillota	Bacilli	*Lysinibacillus*	*macrolides* (sp)	MT197307.1
	Pseudomonadota	Gammaproteobacteria	*Ignatzschineria*	*larvae* (sp.)	HQ696404.1
	Actinomycetota	Actinomycetes	*Georgenia*	*satyanarayanai*	KT720365.1
	Bacillota	Bacilli	*Niallia*	*circulans*	CP053989.1
	Bacillota	Bacilli	*Lysinibacillus*	*fusiformis*	MN795736.1
	Pseudomonadota	Gammaproteobacteria	*Pseudomonas*	sp.	LC185210.1
	Pseudomonadota	Betaproteobacteria	*Bordetella*	sp.	DQ302159.1

## Data Availability

The original data presented in the study are openly available in NCBI Sequence Read Archive (SRA) under the project PRJNA1282731.

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
