# Peer review of "First Attempt at Synthetic Microbial Communities Design for Rearing Gnotobiotic Black Soldier Fly Hermetia illucens (Linnaeus) Larvae"

_insects, 2025, doi:10.3390/insects16080851_

Round 1

Reviewer 1 Report

Comments and Suggestions for Authors
  • I recommend that the authors carefully revise the manuscript for sentence length, complexity, and clarity, ideally with the help of a native English speaker or professional language editor.
  • The Introduction section provides a comprehensive background on BSF microbiota, substrate variation, and probiotic applications. However, it is relatively long and at times overly detailed, especially regarding previously published microbiota studies and general information about organic residue use.

    I recommend that the authors streamline the Introduction to focus more directly on the knowledge gap, study objectives, and hypotheses. Reducing some of the background detail—particularly information already well-established in the literature—would help improve the focus and maintain reader engagement. A more concise Introduction would also make the novelty of the study stand out more clearly.

  • Please write scientific names after common names without parentheses. In title, line 9, and line 45,
  • Please add the author name, order and family name when the scientific name first written. This can be included in the title if there is no word limit. If it is, only the author's name should be included in the title, followed by the author's, order and family names in the abstract( First mention of the scientific name) as specified above.
  • Using 'diet' instead of 'feed' is more accurate in this context, as it better describes the nutritional composition of the material consumed by BSF larvae. Please check  throughout the text.
  • please Improve referencing style consistency and citation placement within sentences
  • Consider discussing the need for verifying whether SynCom members successfully colonized the larval gut (e.g., via qPCR, FISH, or strain-specific sequencing).

  • This is especially important when drawing conclusions about functional effects

  •  
  • The hypotheses regarding the necessity of microbial diversity and dose-dependence are mentioned but not clearly tested or supported; clarify this in the Discussion

Comments on the Quality of English Language

While the manuscript is generally well-structured and the scientific content is clear, the writing style—particularly in the Introduction and Discussion sections—could benefit from revision to enhance clarity and readability.

Many sentences are unnecessarily long and contain multiple clauses, which can obscure the main ideas and make it more difficult for readers to follow the arguments. Shortening sentences and using a more direct structure would improve the overall flow and comprehension of the text.

The sentences, especially in the Introduction section, are excessively long and contain numerous comma-separated subordinate clauses.This could be revised into two or three shorter sentences, each focusing on one core idea.

Reviewer 2 Report

Comments and Suggestions for Authors

Interesting and well-designed research. I only have minor editorial comments and suggestions.

Please replace the word "weight" with mass throughout the manuscript. Please italicize all genus and species names. See lines 175-188, 513, and 518.

Add a comma after all e.g., and i.e.

line 46: BSF larvae are a particularly...

line 59: ...greater economic value than...

line 85: add ")" after host

line 95: cause, not provoke

lines 122-123: ...projected to increase larval growth in a dose-dependent manner.

line 126: is it LAboratoire or Laboratoire?

line 196: change bananas to pineapple

line 199: change roman to romain

line 226: flame sterilized spoon

line 239: saline for solution?

line 258: Petri

line 262: Tukey

line 314: length

lines 318-342: no need to report SD and SE, pick one. Can express and mean ± SE (n) for each treatment

line 373: microbiota

line 376: As expected, the VG SF...

line 386: Bacterial genera...

Figs 22 and 3: x-axis should read: Larval age (days post-hatch)

line 391: The genera are shown...

line 395: (days 4, 8 and 12).

Line 399: BSF

lines: 451-452: ...energy metabolism and biosynthesis of vitamins.

line 507: This study had no way to track the establishment...

Appendix A: What is Scheme 0.?

Figure2 A1, A2, A3. x-axis should be labeled: Larval age (days post-hatch) for each graph

References: italicize Hermetia illucens in all references. Also, line 608.

line 592: ® not ((R))

Comments on the Quality of English Language

Very good English. Only minor suggestions, see above.

Reviewer 3 Report

Comments and Suggestions for Authors

The article is relevant, contains new data and can be published after minor corrections.

Round 2

Reviewer 1 Report

Comments and Suggestions for Authors

Thanks for the corrections.